# Improvement of Biological Effects of Root-Filling Materials for Primary Teeth by Incorporating Sodium Iodide

**DOI:** 10.3390/molecules27092927

**Published:** 2022-05-04

**Authors:** Ji-Myung Choi, Huong Thu Vu, Seong-Jin Shin, Jun-Yong Ahn, You-Jin Kim, Sol Song, Mi-Ran Han, Jun-Haeng Lee, Jong-Soo Kim, Jonathan C. Knowles, Hae-Hyoung Lee, Ji-Sun Shin, Jong-Bin Kim, Jung-Hwan Lee

**Affiliations:** 1Department of Pediatric Dentistry, College of Dentistry, Dankook University, 119 Dandae-ro, Cheonan 31116, Korea; ji3575@gmail.com (J.-M.C.); songsolnara@naver.com (S.S.); miraneee@dankook.ac.kr (M.-R.H.); haeng119@naver.com (J.-H.L.); jskim@dku.edu (J.-S.K.); 2Institute of Tissue Regeneration Engineering (ITREN), Dankook University, 119 Dandae-ro, Cheonan 31116, Korea; huong.vuthudr@gmail.com (H.T.V.); ko2742@naver.com (S.-J.S.); j.knowles@ucl.ac.uk (J.C.K.); haelee@dku.edu (H.-H.L.); 3Department of Biomaterials Science, College of Dentistry, Dankook University, 119 Dandae-ro, Cheonan 31116, Korea; ajy4402@dankook.ac.kr (J.-Y.A.); yujin10316426@gmail.com (Y.-J.K.); 4Department of Nanobiomedical Science & BK21 PLUS NBM Global Research Center for Regenerative Medicine, Dankook University, 119 Dandae-ro, Cheonan 31116, Korea; 5UCL Eastman-Korea Dental Medicine Innovation Centre, Dankook University, 119 Dandae-ro, Cheonan 31116, Korea; 6Cell & Matter Institute, Dankook University, 119 Dandae-ro, Cheonan 31116, Korea; 7Division of Biomaterials and Tissue Engineering, Eastman Dental Institute, Royal Free Hospital, Rowland Hill Street, London NW3 2PF, UK; 8The Discoveries Centre for Regenerative and Precision Medicine, Eastman Dental Institute, University College London, Gower Street, London WC1E 6BT, UK; 9Mechanobiology Dental Medicine Research Center, Dankook University, 119 Dand-ro, Cheonan 31116, Korea

**Keywords:** primary teeth, root-filling material, sodium iodide, iodoform, root canal treatment, root resorption

## Abstract

Therapeutic iodoform (CHI_3_) is commonly used as a root-filling material for primary teeth; however, the side effects of iodoform-containing materials, including early root resorption, have been reported. To overcome this problem, a water-soluble iodide (NaI)-incorporated root-filling material was developed. Calcium hydroxide, silicone oil, and NaI were incorporated in different weight proportions (30:30:X), and the resulting material was denoted DX (D5~D30), indicating the NaI content. As a control, iodoform instead of NaI was incorporated at a ratio of 30:30:30, and the material was denoted I30. The physicochemical (flow, film thickness, radiopacity, viscosity, water absorption, solubility, and ion releases) and biological (cytotoxicity, TRAP, ARS, and analysis of osteoclastic markers) properties were determined. The amount of iodine, sodium, and calcium ion releases and the pH were higher in D30 than I30, and the highest level of unknown extracted molecules was detected in I30. In the cell viability test, all groups except 100% D30 showed no cytotoxicity. In the 50% nontoxic extract, D30 showed decreased osteoclast formation compared with I30. In summary, NaI-incorporated materials showed adequate physicochemical properties and low osteoclast formation compared to their iodoform-counterpart. Thus, NaI-incorporated materials may be used as a substitute for iodoform-counterparts in root-filling materials after further (pre)clinical investigation.

## 1. Introduction

Dental pulp tissue can be damaged due to dental caries, traumatic injuries, or inadequate restorative procedures [1,2]. When pulp is infected, pulpectomy is usually carried out through the mechanical removal of necrotic tissue, irrigation, and filling of the root canal using an antibacterial biomaterial [3]. For the primary teeth, one of the ideal properties of root canal filling materials is to retain the primary tooth until successful permanent tooth eruption by eliminating the infection of the primary tooth [4].

As a root canal filling material for primary teeth, calcium hydroxide has been used as a potent antibacterial agent that disassociates into calcium and hydroxyl ions, leading to elevated pH levels, which inhibit bacterial enzymes and activate tissue enzymes, leading to a mineralizing effect [5]. The addition of silicone oil and iodoform to calcium hydroxide, a commercial product named Vitapex^®^ (Neo Dental Chemical Products Co., Ltd., Tokyo, Japan), was introduced in 1979 by Kawakami et al. Silicone oil acts as an ideal vehicle that allows gradual and slow ionic release in tissues with low solubility [6]. Iodoform (triiodomethane; CHI_3_) is a relatively water-insoluble, lemon-yellow powder that releases iodine, which acts as a strong antiseptic, resulting in high reactivity by precipitating proteins and oxidizing essential enzymes [7]. In addition, iodoform is nonirritating, radiopaque, and resorbable in clinical settings so it has been considered an ideal component of root-filling material of primary teeth [8]. However, several studies have shown that iodoform accelerates root resorption of primary teeth after root canal treatment [9,10,11,12].

Root resorption in primary teeth is a physiological process experienced by erupting permanent teeth [13]. Physiologic root resorption is a scheduled, programmed event, but accidentally accelerated root resorption has also been observed in some cases. It has been documented in cases of inflammation from trauma [14], pulp necrosis [15], and pulpotomy treatment [16] in primary molars. Additionally, root resorption of primary teeth after pulpectomy with iodoform-containing root filling material seems to occur earlier [9,10,11,12]. On these bases, Vitapex^®^ containing iodoform is doubted to hasten the root resorption of primary teeth and, in actuality, early root resorption of primary teeth can often be seen in clinical practice. To overcome this problem, it may be necessary to change iodoform to a different iodine source, and in this study, water-soluble sodium iodide (NaI) was selected as an alternative to water-insoluble iodoform [17]. Like iodoform, NaI is a powder form with a melting point of over 37 °C and has antibacterial effects with the help of iodine (or iodide). It has already been used as an iodine supplement for the treatment of thyroid disease in clinical practice [18].

The purpose of this study was to investigate the physicochemical and biological properties of mixtures of calcium hydroxide, silicone oil, and NaI at different weight ratios compared to iodoform-incorporated paste as a homemade model of Vitapex^®^. The null hypothesis in this study is that sodium iodide-incorporated paste (NaI@P) presents a similar capacity for osteoclast differentiation to iodoform-incorporated paste (Io@P).

## 2. Results

### 2.1. Physicochemical Characteristics

The results of flow, film thickness, viscosity, and radiopacity are shown in Table 1. All groups satisfied the minimum required flowability value of 17 mm, as recommended by the ISO standard (Figure 1a). NaI@P exhibited greater flow properties than Io@P (*p* < 0.001). The film thickness of all groups except D5 was higher than that of I30 (Figure 1b). The viscosity of the groups decreased with increasing frequency (Figure 1c). The flow and viscosity decreased; the film thickness increased as the NaI concentration increased. I30 and D10 presented similar viscosities and film thicknesses among the groups. The ISO standard recommends that radiopacity should not be less than 3 mm of aluminum, and all groups except for D5 conformed to the standards (Figure 1d). I30 presented enhanced superior radiopacity, followed by D30, D20, D10, and D5, and a considerable significant difference was observed among the groups (*p* < 0.001).

The water absorption and solubility results are shown in Table 2. NaI@P absorbed significant water, leading to weight gain, but Io@P showed almost no weight change for 28 days (Figure 1e). NaI@P showed higher solubility than Io@P, except D5 (Figure 1f,g). The water absorption and solubility of D30 were significantly higher than those of I30 (*p* < 0.001).

D5, D30, and I30 were selected for use in the following experiments as representative samples. Through optical imaging, calcium hydroxide and silicone oil were evenly blended with NaI or iodoform. D5 and D30 were blue-green, and I30 was yellowish (Figure 2a). The results of the iodine, sodium, and calcium ion releases and pH are listed in Table 3. The iodine ion was released in NaI@P, whereas it was not detected in I30 (Figure 2b). The sodium and calcium ion release of D30 was the highest, followed by those of D5 and I30, according to ICP/AES analysis (*p* < 0.001). The pH of D30 was also higher than that of I30 (*p* < 0.001) (Figure 2c). The chromatogram obtained by LC/MS in positive mode is presented in Figure 2d. A noticeable difference in the intensity of I30 compared to D5 and D30 appeared in the peak at a retention time of approximately 11.949 min. Analysis of the full-scan mass spectrum at 11.949 min showed that the molecular ion with a mass of 610.184 ${\left[M+H\right]}^{+}$ was the major ion. As the calculated molecular weight was 610.184, the predicted formula was $C_{18}H_{49}N_{5}OS_{3}Si_{5}$, and there was no result from annotation sources. Therefore, this molecule was highly detected in I30 compared to D5 and D30.

### 2.2. Biological Characteristics

#### 2.2.1. Cytotoxicity Test

The results of the cell viability of the material extracts at different concentrations (control, 12.5%, 25%, 50%, and 100%) are shown in Table 4 and Figure 3a. The cytotoxicity of the materials against RAW 246.7 cells using CCK-8 was reexamined by fluorescence microscopy images after Live/Dead cell staining (Figure 3b). In the I30 and D5 extract groups, there were no significant differences in cell viability at different percentages of extracts, except at the 100% extract, in which there was a slight increase in cell proliferation compared to the 0% extract. In the D30 extract groups, the cell viability of the 12.5%, 25%, and 50% extracts was not different from that of the control group, while it significantly decreased in the 100% extract group. Cell images were obtained after Live/Dead cell staining to confirm cell survival. Live cells were stained green, and dead cells were stained red. The images of the stained cells showed fewer live cells in the 100% extract of the D30 group than in the other groups, which was consistent with the cell viability assay results. From the CCK-8 and Live/Dead cell staining results, it was generally assumed that the approximately 50% extract of the I30, D5, and D30 materials does not induce toxicity to the culture of RAW 264.7 cells. Therefore, this concentration of the extracts was selected for use in subsequent experiments.

#### 2.2.2. Tartrate-Resistant Acid Phosphatase (TRAP) Staining and Actin Ring Staining (ARS)

The effects of the materials on osteoclast formation were measured using TRAP staining and ARS, which have been widely used as biomarkers of osteoclasts. RAW cells were cultured in osteoclast differentiation media for 5 days and then stained for TRAP or actin rings (Figure 4a,b). The mean values and the standard deviation of number and size of osteoclasts from TRAP and actin ring staining are listed in Table 5. Figure 4a shows osteoclast formation from TRAP staining under the condition of 50% extract of materials (I30, D5, and D30) or without the extract (control group). Only the cells with more than 3 nuclei (positive TRAP) were counted, and the areas were measured. The size and number of monomorphic osteoclasts increased significantly in the I30 and D5 groups compared to the control group (*p* < 0.05). The extract of I30 induced the highest number of osteoclasts (500 ± 89.15), followed by the extract of D5 (397 ± 47.62). When treated with 50% extract of D30, RAW cells were less differentiated into osteoclasts than in the control groups, with the lowest number of differentiated cells (122 ± 48.38); however, the difference was not significant (Figure 4c). As seen in an actin ring formation, as shown in Figure 4b, stimulation with the extracts of I30 and D5 increased the size of the actin rings, while the size of the actin rings in the D30 and control groups did not differ.

#### 2.2.3. Quantitative Polymerase Chain Reaction (qPCR)

To verify the effects of the material-extracted solutions on the expression of genes related to osteoclast differentiation from RAW 264.7 cells, the mRNA expression levels were evaluated by qPCR at 3-time points (days 2, 3, and 4) and the results are shown in Table 6, Figure 4d. As seen in most groups, NFATc1, a key modulator of osteoclast maturation, was upregulated starting from day 2 after treatment. C-Fos, which regulates NFATc1, was also detected from day 2 and increased significantly from day 3. TRAP mRNA was upregulated on day 2, followed by a further significant increase on day 3 (58-fold) and a drastic decline on day 4. The expression of cathepsin K remained high on days 2, 3, and 4. The extracts of I30 and D5 were demonstrated to significantly upregulate the expression during different periods in comparison with the control group. The extract of D30 generally tended to show similar or slightly lower mRNA expression levels than the control group.

Collectively, these results indicate that D30 showed decreased osteoclast formation compared to I30 and D5 in an in vitro study.

## 3. Discussion

The null hypothesis that the addition of sodium iodide instead of iodoform from model homemade root-filling material with a similar chemical composition to Vitapex^®^ would show a similar degree of osteoclast differentiation was rejected. Extract from physical property-optimized NaI@P at a proportion of 3:3:3 exhibited less osteoclast differentiation capacity than its Io@P counterpart at nontoxic concentrations.

Vitapex^®^, a commercial product, consists of iodoform (40.4%), calcium hydroxide (30.3%), silicone oil (22.4%), and others (6.9%). In this study, it was impossible to make a perfect counterpart of Vitapex^®^ instead of NaI@P since the component of “others (6.9%)” in Vitapex^®^ is not indicated specifically by the manufacturing company. Thus, there was no choice but to make a homemade version of Vitapex^®^ (Io@P), which consisted of calcium hydroxide, silicone oil, and iodoform, and NaI@P in a different proportion to compare the effects of iodoform and NaI itself. The proportion of the components in NaI@P and Io@P in this study was simplified for optimization.

The physical properties of the root-filling material strongly affect clinical performance. The flowability, film thickness, and viscosity of root canal filling materials are crucial factors to assess filling ability. The flow of canal filling material is important to reach accessory canals and dentinal wall irregularities in order to increase sealing capacity [19]. The thinner the film thickness and the higher the flowability, the greater the possibility of material penetration into the accessory root canal [20]. The space between the filling material and the canal wall provides an opportunity for microorganisms to reinfect from the coronal and apical sides [21]. An appropriate flow that is not excessive is also required because excessive flow increases the chance of material extrusion toward the periapical region [22]. Viscosity is a quantitative parameter for the evaluation of the rheological properties of the filling material and may help to achieve an ideal flow pattern [19]. According to findings of this study, the tested pastes are pseudoplastic, which means that their viscosity decreases as the shear rate increases [21]. In the present study, as the concentration of NaI increases, the flowability decreases, and the film thickness and viscosity increase. The difference in the liquid/powder ratio of the pastes may have influenced the results [23]. The flow of all groups meets the ISO standards, and the film thickness and viscosity of D10 are comparable to those of I30. The film thickness of all groups did not fulfill the ISO standard, which recommends less than 50 μm as root-filling materials, and the suspected reason is that all experimental pastes were mixed in the laboratory by an experimenter, rather than being a ready-made product.

Radiopacity is also an important requirement for root-filling materials. Root filling materials must be radiopaque to enable the evaluation of filling quality and the position in the root canal, as well as the correction of filling [20]. In the present study, all samples except D5 were found to satisfy the ISO standard recommendation, in which the radiopacity value should be higher than 3 mmAl. The Io@P demonstrated better radiopacity results. Iodine is a radiopaque substrate. The molecular weight of NaI is 149.89 g/mol and that of iodoform (CHI_3_) is 393.732 g/mol, so the number of iodine moles per weight is 1 × 1/149.89 and 3 × 1/393.732, respectively. Therefore, Io@P presents better radiopacity.

NaI@P showed higher water absorption and solubility than Io@P in the study. The difference lies in the degree of affinity of the component with water. NaI is hydrophilic, but iodoform is a hydrophobic material, and the water solubility is 1842 g/L at 25 °C and 100 mg/L. The volumetric expansion due to water absorption is thought to be beneficial to fill the space of the root canals. However, too much expansion compromises the tooth structure integrity; thus, further investigation is needed. Solubility is an important factor of root-filling materials, and low solubility is desirable for root-filling material not to be extruded and irritate periapical tissue [24]. NaI@P, especially D30, showed high solubility when supposed to moisture because its water-soluble property, which exceeded the recommended maximum mass loss of 3% in ISO 6876; therefore, further supplemental material development is needed to reduce the solubility. The iodine ion was not detected in I30 because it is water-insoluble and is not well ionized. Meanwhile, NaI@P disclosed the release of iodine ion as tens to thousands of ppm, supporting water-soluble characteristic of NaI and possibly affecting antimicrobial effects by iodine ions. Compared to antimicrobial effects by iodoform (CHI_3_), NaI can have different antimicrobial effects and underlying mechanisms by the direct effects of the iodine ions released. Calcium ion release allows for the reduction in permeability of new capillaries in granulated tissue of depulped teeth, which has a synergistic effect with hydroxyl ions activating alkaline phosphatase favoring mineralization [5]. Its release in tissue has also been reported to activate Ca-dependent adenosine triphosphatase, which may react with tissue calcium dioxide to form calcium carbonate crystals, leading to crystallization of apatite into voids and spaces to improve sealing capacity [22]. This process is also necessary for cell differentiation and migration [25]. NaI@P released more calcium than Io@P, allowing for more mineralization and sealing effects in the tissue. As calcium hydroxide is ionized into calcium and hydroxyl ions, the pH (hydroxyl ion release) is also higher in D30 than in I30, and is suspected to have more antibacterial and mineralizing effects in D30 [5,26,27].

Toxicity assays revealed that I30 has no cytotoxicity, which is consistent with previous experimental results using commercial material (Vitapex^®^). The study performed by Hung et al. revealed that the Ca(OH)_2_
+iodoform group and Vitapex^®^ group showed high survival rates on human osteosarcoma cell lines [28]. Similarly, Vitapex^®^ resulted in high cell viability, the concentration of the eluate did not affect fibroblast viability, and the cell viability was slightly higher at high concentrations but not significantly different [29]. Another study confirmed that low concentrations of iodoform-containing materials (Endoflas^®^) induced macrophage proliferation; however, when macrophages were cultured at a high concentration, they decreased the viability of macrophages by 70%, and the expected reason for cell death could be the zinc oxide eugenol and paramonochlorophenol liquid in their tested material. To our knowledge, this is the first study examining the effect of NaI as a component of root-filling material on osteoclast precursor (RAW 264.7 cells). The survival rate at concentrations 50% or lower than 50% showed no differences, but as a limitation, the cell viability decreased in the 100% of the D30 extracted solution. In this study, the D5 extract did not affect cell survival; in particular, cell viability increased. Cell viability depends on the concentration of the D30 extracted solution.

The degree of osteoclast differentiation of RAW 264.7 cells treated with I30, D5, and D30 was evaluated via TRAP staining, actin ring formation, and the expression of osteoclastogenesis-related genes. A 50% concentration of media that showed noncytotoxic properties according to Live/Dead staining and CCK-8 assay tests was adopted. The formation and activity of TRAP-positive cells are well-known ways to determine osteoclast development and function [30]. TRAP is a highly expressed enzyme in osteoclasts and is localized within the ruffled border [31]. In the present study, D30 induced diminished osteoclast formation and TRAP activity compared with I30 and D5. Actin ring staining was used to determine the structure and function of the cytoskeleton of osteoclasts. The actin cytoskeleton, which is a component of osteoclasts, engages in the activity required for resorption and is dynamic, undergoing rounds of resorption with actin ring formation and movement without actin ring formation [32]. When stimulated for resorption, osteoclasts organize disordered, diffused microfilaments into a belt or ring-like structure of elaborate actin rings, and the actin rings surround a ruffled plasma membrane that is involved in resorption [33,34]. From the actin structure visualized by fluorescent phalloidin after fixation with paraformaldehyde, the extracts of I30 and D5 induced an increase in actin rings compared to the D30 extract, indicating the ability of cells to resorb teeth and bone.

The qPCR results indicated the effect of the materials on osteoclastogenesis through the regulation of key transcription factors, such as NFATc1 and c-Fos (Figure 4d). NFATc1 and c-Fos are important transcription factors for RANKL-mediated osteoclast differentiation, fusion, and activation, and the recruitment and subsequent stimulation of c-Fos results in the activation of NFATc1 [35]. It has also been noted that overexpression of NFATc1 and c-Fos was observed in I30 and D5, whereas D30 exerted a relative inhibitory effect on these transcript genes. Moreover, the release of osteoclastogenesis-related genes such as TRAP and cathepsin K, which are the main markers responsible for the degradation of bone mineral and collagen matrices, was upregulated in the I30 and D5 extracts but downregulated in the D30 extract solution [36]. These data indicate that D30 shows lower osteoclastogenesis than I30 and D5.

The relationship between root canal treatment and primary tooth root resorption has not yet been well established. The dental follicle of the permanent successor is believed to play a critical role in primary tooth resorption, providing an advantageous environment for the recruitment of mononuclear cells and differentiation of osteoclasts [37,38,39]. Cells from the stellate reticulum of erupting teeth secrete a parathyroid hormone-related protein (PTHrP) and interleukin-1α, which are regulatory molecules required for tooth eruption, leading to stimulation of the dental follicle cells. These cells secrete monocyte recruiting factors such as colony-stimulating factor-1 (CSF-1), leading to the fusion of monocytes and macrophages to form osteoclasts [40]. This study focused on the degree of differentiation of RAW “macrophages” cells into osteoclasts when exposed to different components and concentrations of root-filling materials through experiments of TRAP staining, actin ring staining, and m-RNA (TRAP, c-FOS, NFATc1 and cathepsin K) expression of q-PCR.

Direct or indirect exposure to low concentrations of iodoform-containing root canal filling material induced the proliferation of macrophages, which are immune cells that activate the inflammatory system and secrete cytokines, such as interleukin 6 and TNF-α, that mediate periapical inflammation and bone resorption [11]. Contact with degraded low concentrations of root canal filling materials in periapical tissue induces the inflammatory macrophage response and root resorption by osteoclasts occurs [41]. Root-filling materials that contain irritating substances can initiate a foreign body reaction directly or indirectly at the periapex, leading to the local formation of multinucleated giant cells, which are considered osteoclasts when specifically in bone tissue [42]. Based on this evidence, it is assumed that an irritating substance in Io@P could stimulate the differentiation of osteoclasts from macrophages directly or indirectly, leading to early root resorption. According to the LC/MS results in this study, unknown molecules presented high intensity in I30 compared with D5 and D30, and $C_{18}H_{49}N_{5}OS_{3}Si_{5}$ is the suspected molecular formula based on molecular weight and components and it is thought to affect osteoclast differentiation; however, in this part, further detailed investigation is needed.

A periradicular lesion from infection or necrosis can affect periradicular tissue directly or indirectly, depending on the continuity and intensity of the stimulus, causing the formation of osteoclasts and external root resorption [43]. Bacteria can induce osteoclastogenesis through a RANKL-dependent pathway or the differentiation of drawn leukocytes into osteoclasts [44]. Moskovitz et al. found that eight (3.3%) of 242 primary molars treated with an iodoform-containing material, Endoflas^®^ (Sanlor & Cia. S. en C.S., Cali, Colombia), presented a new radiolucent defect or enlargement of the existing periapical radiolucency, indicating that one of the components of root canal filling material can evoke cyst-like radiolucent defects, leading to early root resorption [10]. They also found accelerated root resorption compared with the degree of root resorption in homologous teeth after root canal treatment with a material composed of iodoform (Endoflas^®^) [9]. It is thought that the accelerated root resorption may be due to irritation by the root-filling materials of the dental follicle, which is the initiation point for root resorption of primary teeth. However, this study observed the degree to which macrophages differentiate into osteoclasts directly, not the effect of the root-filling material on the dental follicle itself.

The present study demonstrated that D30 was superior to I30 and D5 in terms of the low differentiation of osteoclasts from RAW cells. One of the limitations is that this study was an in vitro biological study, so that the complexities of biological responses, including the effect on dental follicles, cannot be adequately assessed by a single in vitro test. Thus, further experiments on animals should be performed prior to commercial use in actual clinical practice to evaluate the full impact and adverse effects of the new materials in vivo. In addition, since the periapical area is a rich cell region, further experiments should be performed to better evaluate the toxicity of D30 on other types of cells.

## 4. Materials and Methods

### 4.1. Preparation of Samples

NaI was ground to make a smaller particle using a planetary ball mill (PM 100 CM, Retsch Co., Haan, Germany), and then the particle was strained into a 100 μm bowl (Figure 5a). The particle size of the strained NaI and existing iodoform was analyzed by a particle size analyzer (PSA) (Horiba LA-950 V2, Beckman, Urbana, IL, USA). The particle size of the strained NaI was similar to that of iodoform, approximately 90 μm, so the same experimental conditions for both materials were applied (Figure 5b). Calcium hydroxide (Sigma–Aldrich, Burlington, MA, USA), silicone oil (Sylgard 184, Dow Corning Co., Midland, MI, USA), and strained NaI (Sigma–Aldrich, Burlington, MA, USA) were blended, and the resulting materials were denoted D5 (30:30:5), D10 (30:30:10), D20 (30:30:20), and D30 (30:30:30) (ratio = calcium hydroxide: silicone oil: NaI) (Figure 5c). Iodoform (Alfa Aesar, Heysham, UK) instead of NaI was incorporated at a ratio of 30:30:30, and the resulting materials were denoted I30.

### 4.2. Physicochemical Characteristics of Samples

Using these prepared pastes, the physicochemical and biological properties were examined. The physicochemical properties were determined according to International Organization for Standardization (ISO) 6876:2012 (Dentistry—Root Canal Sealing Materials). All tests were performed at room temperature (23 ± 2 °C) and 5% relative humidity.

#### 4.2.1. Flow

Each 0.05 mL sample was placed in the center of a glass plate with a size of 40.0 × 40.0 × 5.0 mm. After 3 min, another glass plate of the same size was placed on top of the first plate and then pressed with a total of 120 g of pressure, including a 100 g weight. This setup remained stable for 7 min. After that, the smallest and largest diameters of each specimen were measured using a digital caliper (Mitutoyo, Kawasaki, Japan), and the mean of the two diameters (in mm) was calculated. If the two diameters were not within 1 mm of each other, the test was repeated. This test was performed ten times and the mean and standard deviation were calculated.

#### 4.2.2. Film Thickness

The thickness of the two stacked glass plates, each 15.0 × 15.0 × 5.0 mm in size, was measured using a micrometer apparatus (Mitutoyo). Each 0.01 mL sample was placed on the center of a glass plate, and another glass plate was placed on top of the first plate. After 3 min, the glass plate was pressed with a 150 N force. At this time, the sample filled the area of the glass plate. Ten minutes after the start of mixing, the thickness of the sample and the two glass plates was measured using a micrometer apparatus. The film thickness was calculated as the difference in thickness with and without the sample between the glass plates. The film thickness (%) is presented as the film thickness of different groups/film thickness of I30 × 100%. The test was repeated ten times and the mean and standard deviation were calculated.

#### 4.2.3. Viscosity

Viscosity measurements were executed on Discovery HR-1 (TRIOS, TA Instruments, New Castle, DE, USA). The instrument was calibrated, a geometry was mounted and calibrated, and then a sample was loaded onto the Discovery HR-1 plate. The experiment progressed, and a graph was plotted in which the x-axis is the frequency (Hz), and the y-axis is the complex viscosity (Pa·s).

#### 4.2.4. Radiopacity

Stainless steel ring molds (diameter, 10.0 mm height, 1.0 mm) were placed over a thin glass plate and filled with the sample. Another thin glass plate was placed over the rings to ensure a sample thickness of 1.0 mm. To classify the radiopacity, an aluminum step wedge graduated 1~9 mm in 1 mm increments was used. Molds filled with the sample were placed next to the aluminum step wedge, and a digital X-ray device (Kodak-2200, Kodak Insight, Rochester, NY, USA) and digital sensor (Kodak Insight) were used to obtain digital radiographs of each specimen. The digital sensor voltage, electric currents, distance, and radiation time were 70 kV, 7 mA, 300 mm, and 0.3 s, respectively. The obtained images were analyzed using ImageJ software version 1.53a (National Institutes of Health, Bethesda, MD, USA). The gray value of the samples was compared with that of 10 mm of aluminum. The test was repeated ten times. The mean gray value for each sample was calculated and converted to millimeters of aluminum (mmAl) by using the calculation formula of Húngaro Duarte et al. [45].

#### 4.2.5. Water Absorption

One gram of each sample (D5, D10, D20, D30, and I30) was placed in the five glass bottles after weighing empty bottles, and 5 mL of PBS solution was added to each. The glass bottles were placed in a shaking incubator (HB-201SF, Hanbaek Scientific Co., Bucheon, Korea) at 220 rpm and 37 °C, and then were weighed after 1, 3, 7, 14, and 28 days. After weighing the glass bottles after 1, 3, 7, 14, and 28 days, all the PBS solution was removed by using a pipette, and then it was placed in a dry oven (WiseVen, Daihan Scientific, Wonju, Korea) at 37 °C for 20 min. After that, the weight of the sample alone was measured. Then, 5 mL of PBS solution was added using a pipette and placed in a shaking incubator again. The water absorption at each time point was calculated by the percentage of weight change. The water absorption at each time *t* was calculated as follows:$$\mathrm{Water}\;\mathrm{absorption}=\frac{\mathrm{Wet}\;\mathrm{weight}\;\mathrm{at}\;\mathrm{time}\;t\;-\;\mathrm{initial}\;\mathrm{weight}}{\mathrm{Initial}\;\mathrm{weight}}\;\times \;100\;(\%)$$

#### 4.2.6. Solubility

To measure the solubility, the experiments were carried out in two different ways.

##### Solubility from Specimen

One gram of each sample was placed in the five glass bottles after weighing empty bottles. Five milliliters of PBS solution were added to each, and the bottles were placed in a shaking incubator (HB-201SF) at 220 rpm and 37 °C and were weighed after 3 days. The PBS solution was removed using a pipette and placed in a dry oven (WiseVen) at 70 °C for 12 h for complete evaporation. After that, the bottles were cooled to room temperature, and then the weight of the bottles was measured. The weight of elution is the difference between the initial weight of the bottle with the sample and its final weight. This test was performed three times and the mean and standard deviation were calculated. The solubility was calculated as follows:$$\mathrm{Solubility}=\frac{\mathrm{Initial}\;\mathrm{weight}-\mathrm{dry}\;\mathrm{weight}}{\mathrm{Initial}\;\mathrm{weight}\;}\;\times \;100\;(\%)$$

##### Solubility from Extract

A Teflon (polytetrafluroethlyene; DuPont, HABIA, Knivsta, Sweden) mold with an internal diameter of 20 ± 1 mm and a height of 1.5 ± 0.1 mm placed on a glass plate was weighed to the nearest 0.001 g ($m_{0})$. The sample was placed in a mold with slight excess and the excess was removed with another glass plate. Then, the process was repeated ($m_{1})$. The weight of the sample was calculated by subtracting the weight of $M_{0}\;$from $M_{1}$. Two specimens were used for each tested sample. The two specimens were immersed in 50 mL distilled water in a shallow dish for the entire surface of the sample to come into contact with the distilled water. The specimens were placed under 100% relative humidity at 37 °C for 24 h. After 24 h, two specimens were removed from a dish, and eluted distilled water was poured into a funnel in a beaker. The beaker with the collected water was placed in an oven at 110 °C. After 6 h, the collected water was evaporated completely without boiling, and the beaker was cooled to room temperature before each weighing. The difference between the original weight of the beaker and its final weight was calculated. This test was performed three times, and the mean and standard deviation were calculated. The solubility was calculated as follows:$$\mathrm{Solubility}=\frac{\mathrm{Final}\;\mathrm{weight}\;\mathrm{of}\;\mathrm{beaker}-\mathrm{original}\;\mathrm{weight}\;\mathrm{of}\;\mathrm{beaker}}{\mathrm{Weight}\;\mathrm{of}\;\mathrm{sample}\;(m_{1}-m_{0})}\;\times \;100\;(\%)$$

#### 4.2.7. Optical Images of D5, D30, and I30

D5, D30, and I30 were selected for use in subsequent experiments as representative samples. To visualize the surface properties of the samples, optical microscopy images of D5, D30, and I30 were acquired using an optical microscope (S39A, Microscopes Inc., Saint Louis, MO, USA).

#### 4.2.8. Extraction Analysis of D5, D30, and I30

Iodide ion release was analyzed by ion chromatography (IC, Metrohm 833 Basic IC plus) with a Metrosep A Supp4 column (250 mm× 4.0 mm) (Metrohm, Herisau, Switzerland), an eluent of 1.8 mM $Na_{2}CO_{3}$/1.7 mM $Na_{2}HCO_{3}$ in distilled water.

Inductively coupled plasma atomic emission spectrometry (ICP/AES) (Optima 8300, Perkin-Elmer, Waltham, MA, USA) was used to determine the concentrations of sodium and calcium ions in each specimen. The ICP/AES settings included a spectral range of 167~782 nm and a detection limit of 10 ppb. The pH of a 2 g/mL sample in distilled water was measured using a pH meter (Orion VERSA STAR Pro, Thermo Fisher Scientific, Waltham, MA, USA). Each sample in the ICP/AES and pH tests was analyzed three times.

Liquid chromatography/mass spectrometry (LC/MS) is a useful, specific, and sensitive analytical technique for the identification of the specific compounds eluted. Liquid chromatography (LC) separates high-molecular-weight compounds by passing them through a chromatographic column so that they emerge sequentially [46]. LC separation methods are combined with MS for sensitive detection and mass measurement of the fragmented ions [47]. The prepared samples of 100 μg/mL were separated with an Ultimate 3000 RSLC (Thermo Scientific). An Acquity UPLC BEH C18 column (2.1 mm × 100 mm, 1.7 μm) was used as the capillary column for LC separation with two mobile phases: mobile phase A was 0.1% formic acid in DW, and mobile phase B was 0.1% formic acid in acetonitrile. Chromatography was run by linear gradient elution; the analysis was initiated with 5% B for 10 min, 100% B for 3 min, and 5% B for 2 min. The flow rate was 0.4 mL/min, and the injection volume was 5 μL. LC was coupled with a Q-Exactive Orbitrap Plus Mass Spectrometer (Thermo Fisher Scientific) in positive mode. The mass parameters were as follows: spray voltage, 3500 V; capillary temperature, 263 °C; full-scan resolution, 70,000 FWHM; collision energy, 30 V.

### 4.3. Biological Characteristics of D5, D30, and I30

#### 4.3.1. In Vitro Test

Osteoclasts are multinuclear giant cells derived from monocyte-macrophage lineage cells. RAW 264.7 cells have been widely used to build models of osteoclast differentiation. Murine RAW 246.7 cells (ATCC, CRL-2593^TM^) were cultured in extracted media to determine the effect of materials on osteoclastogenesis. Two grams of material containing 10 mL α-MEM (alpha-minimal essential medium; Welgen, Gyeongsan, Korea) was incubated at 37 °C for 24 h. After that, the media was collected and passed through a 0.22 µm filter (Corning, Corning, NY, USA) to eliminate contamination. The extracted media was obtained to test for cytotoxicity and bioactivity properties. For use in the cytotoxicity and bioactivity experiments, the extracted solution was diluted at different ratios (0~100%) in α-MEM, and then diluted media was prepared with the addition of 10% heated in-active fetal bovine serum (FBS; Corning) and 1% penicillin/streptomycin (PS; Thermo Fisher Scientific).

#### 4.3.2. Cytotoxicity Test

To understand the effects of the material extract on osteoclast differentiation, a nontoxic concentration of the extract was determined and then used in further osteoclast differentiation. The toxicity of the extracted materials on RAW 264.7 cells was based on cell viability using a cell counting kit-8 (CCK-8; Dojindo, Kumamoto, Japan) assay. In brief, RAW 264.7 cells were seeded in an uncoated 96-well plate (SPL Life Sciences, Gyeonggi, Korea) at a density of 2 × 10^4^ cells/well (α-MEM, 10% FBS, and 1% PS) and incubated overnight. Then, cells cultured in media were shifted to extracted solutions at different concentrations for the next 24 h, and the 0% extracted group was used as a negative control group. After 24 h, the medium containing 10% CCK-8 solution was replaced in each well of the plate, followed by incubation for 2 h at 37 °C. The absorbance of each well at 450 nm was recorded using a microplate reader (${\mathrm{Varioskan}}^{\mathrm{TM}\;}$ LUX, Thermo Fisher Scientific). Cell viability (%) was quantified by the average absorbance of different groups/average absorbance of the negative control group × 100%. Cell survival was also examined by Live/Dead cell staining (0.5 µM calcein AM and 2 µM ethidium homodimer-1 solutions, Thermo Fisher Scientific) [32,48], and images were taken using an optical microscope (Olympus IX71; Olympus, Tokyo, Japan).

#### 4.3.3. Tartrate-Resistant Acid Phosphatase (TRAP) and Actin Ring Staining (ARS)

For osteoclast differentiation, RAW 264.7 cells were seeded at a density of 4 × 10^4^ cells/well in a 12-well plate (SPL Life Sciences) in α-MEM with 10% FBS and 1% PS for 24 h. After that, the cells were treated with noncytotoxic diluted extracts and in the presence of osteoclast differentiation supplement (50 ng/mL RANKL (Peprotech, Middlesex County, NJ, USA), 10 ng/mL M-CSF (Peprotech), and 1 ng/mL TGF-$\beta_{2}$ (R&D Systems, Minneapolis, MN, USA)). The media were refreshed every 2 days. After 5 days of culture, the number of differentiated osteoclasts was determined by tartrate-resistant acid phosphatase staining (TRAP Staining Kit, Cosmo Bio, Tokyo, Japan) [49]. TRAP-positive multinuclear (>3 nuclei) cells (TRAP(+)) were regarded as osteoclasts. The cells were washed three times with phosphate-buffered saline (PBS) and fixed with 4% paraformaldehyde (PFA, Tech & Innovation, Gangwon, Korea) for 15 min at room temperature. The fixed cells were washed 3 times with PBS (Tech & Innovation). One vial of the chromogenic substrate was dissolved in 5 mL of tartrate-containing buffer. Then, 250 μL of dissolved chromogenic substrate was added to each well and incubated at 37 °C for 60 min. After that, the stained cells were washed with deionized water and observed using an optical microscope (IX71, Olympus). The cells were fixed with 4% PFA for 15 min at room temperature. The fixed cells were washed three times with PBS and permeabilized with 0.5% Triton X-100 (Sigma–Aldrich) for 10 min. After washing 3 times with PBS, the cells were stained with Alexa Fluor^®^ 546 phalloidin (Thermo Fisher Scientific) for 30 min, and then DAPI (4′,6-diamidino-2-phenylindole, Thermo Fisher Scientific) for 10 min. The size of the actin rings was manually drawn and measured using ImageJ software version 1.53a (National Institutes of Health)**.**

#### 4.3.4. Quantitative Polymerase Chain Reaction (qPCR)

RAW cells were differentiated into 3 groups of extracted solution for 2, 3, and 4 days. Total RNA was isolated using Ribospin^TM^ (cat. no. 304-150; *GeneAll* Biotechnology, Seoul, Korea). According to the manufacturer’s instructions, the RNA concentration was measured using a Nanodrop (Thermo Fisher Scientific). The gene expression levels of cathepsin K, TRAP, nuclear factor-activated T cells c1 (NFATc1), and c-Fos, which are relevant to osteoclast differentiation, were determined. Glyceraldehyde-3-phosphate dehydrogenase (GAPDH) served as the housekeeping gene. The primer sequences were as follows: cathepsin K forward 5′-GAAGAAGACTCACCAGAAGCAG-3′, reverse 5′-TCCAGGTTATGGGCAGAGATT-3′; TRAP forward 5′-ACTTCCCCAGCCCTTACTACCG-3′, reverse 5′-TCAGCACATATGCCCACACCG-3′; NFATc1 forward 5′-GGAGCGGAGAAACTTTGCG-3′, reverse 5′-GTGACACTAGGGGACACATAACT-3′; c-Fos forward 5′-CGGGTTTCAACGCCGACTA-3′, reverse 5′-TTGGCACTAGAGACGGACAGA-3′; and GAPDH forward 5′-ACTTTGTCAAGCTCATTTCC-3′, reverse 5′-TGCAGCGAACTTTATTGATG-3′. Undifferentiated RAW cells were used as a control throughout the study. The results are expressed as relative mRNA expression [50].

### 4.4. Statistical Analysis

The data are presented as the means ± standard deviations (SDs) and were analyzed using SPSS software 21.0 (SPSS Inc., Chicago, IL, USA). To analyze the significant differences in physicochemical properties, one-way analysis of variance (ANOVA) and Tukey’s honest significant difference (HSD) test were performed. The statistical significance of the cytotoxicity test was calculated using one-way ANOVA followed by a two-sided Dunnett’s multiple comparison test. The statistical significance of TRAP staining and qPCR experiments was calculated using one-way ANOVA followed by a two-sided Tukey’s multiple comparison test. The significance level adopted was *p* < 0.05.

## 5. Conclusions

NaI-incorporated root-filling material diminished osteoclast differentiation compared with iodoform-incorporated material. Additionally, NaI-incorporated root-filling material satisfied physicochemical properties; therefore, NaI is a promising material as a substitute for iodoform. Furthermore, with extensive experiments including clinical trials and optimization of materials characteristics, NaI-incorporated root-filling material could be substitute of the commercially available product Vitapex^®^ in the future.

## Figures and Tables

**Figure 1 molecules-27-02927-f001:**
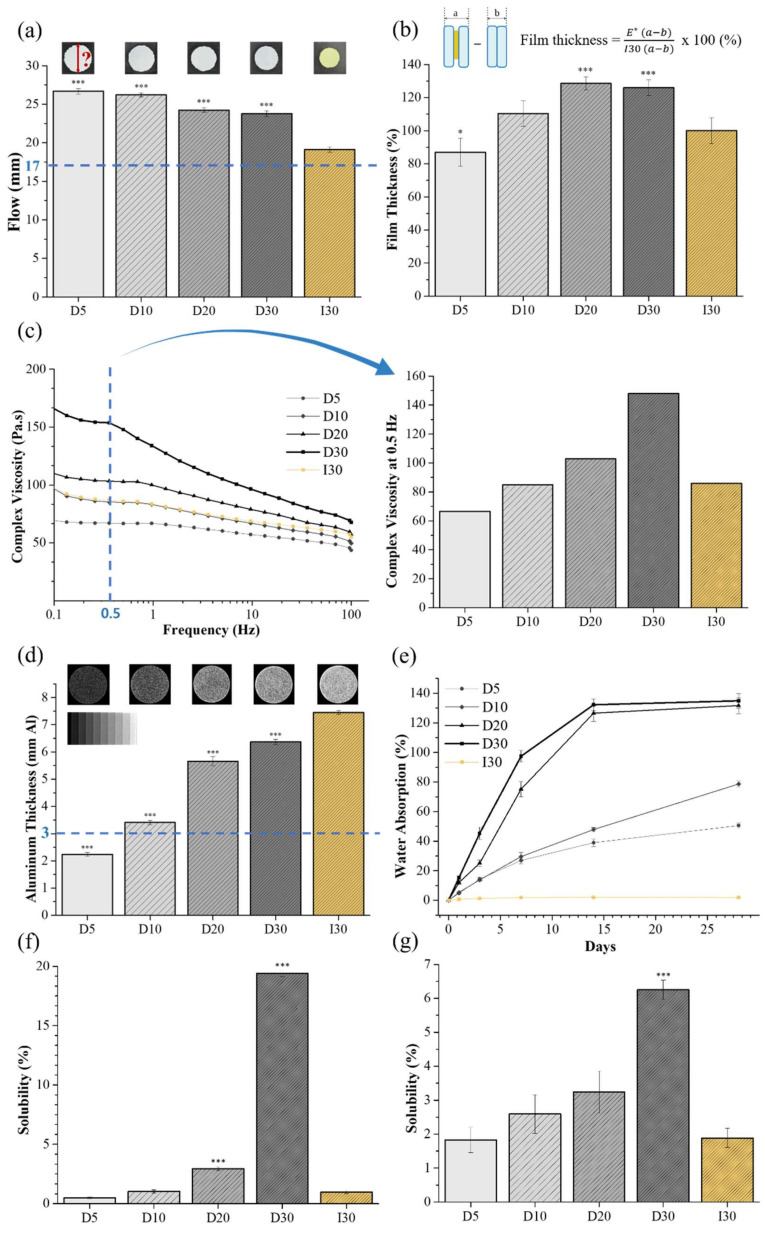
Physical properties of NaI-incorporated paste (NaI@P). (**a**) Flowability. (**b**) Film thickness. (**c**) Viscosity. (**d**) Radiopacity. (**e**) Water absorption by weight change over 28 days. (**f**) Solubility from specimen. (**g**) Solubility from extract. Statistical significance was calculated using a one-way analysis of variance (ANOVA) followed by Tukey’s honestly significant difference compared to the control (I3O). * *p* < 0.05, *** *p* < 0.001.

**Figure 2 molecules-27-02927-f002:**
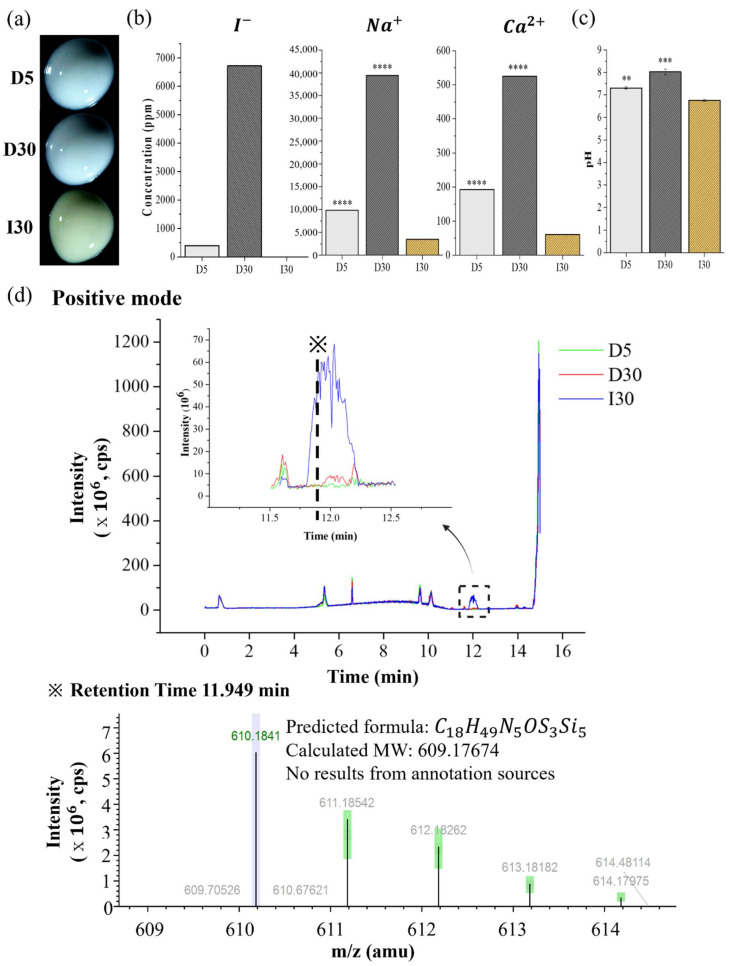
Optical images and extraction analysis of NaI-incorporated paste (NaI@P). (**a**) Optical images. (**b**) Iodine, sodium, and calcium ion releases. (**c**) The pH. (**d**) Molecules extracted by liquid chromatography/mass spectrometry (LC/MS) (*m*/*z* = mass to charge ratio). Statistical significance was calculated using one-way analysis of variance (ANOVA) followed by Tukey’s honest significant difference compared to the control (I30). ** *p* < 0.01, *** *p* < 0.001, **** *p* < 0.0001.

**Figure 3 molecules-27-02927-f003:**
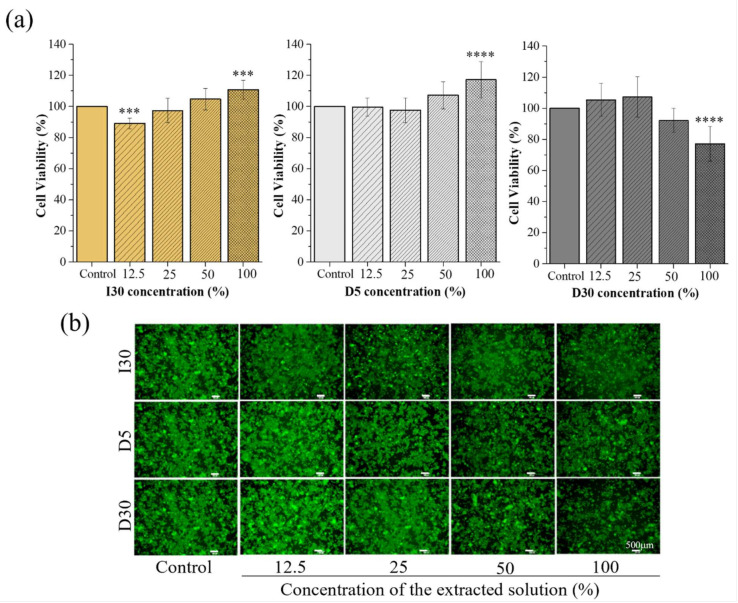
Cytotoxicity of the extract of NaI-incorporated paste (NaI@P). (**a**) Cell viability results from the CCK-8 assay. (**b**) Live/Dead staining results. The statistical significance of (**a**) was calculated using a one-way analysis of variance (ANOVA) followed by a two-sided Dunnett’s multiple comparison test compared to the control. *** *p* < 0.001, **** *p* < 0.0001, *n* = 6.

**Figure 4 molecules-27-02927-f004:**
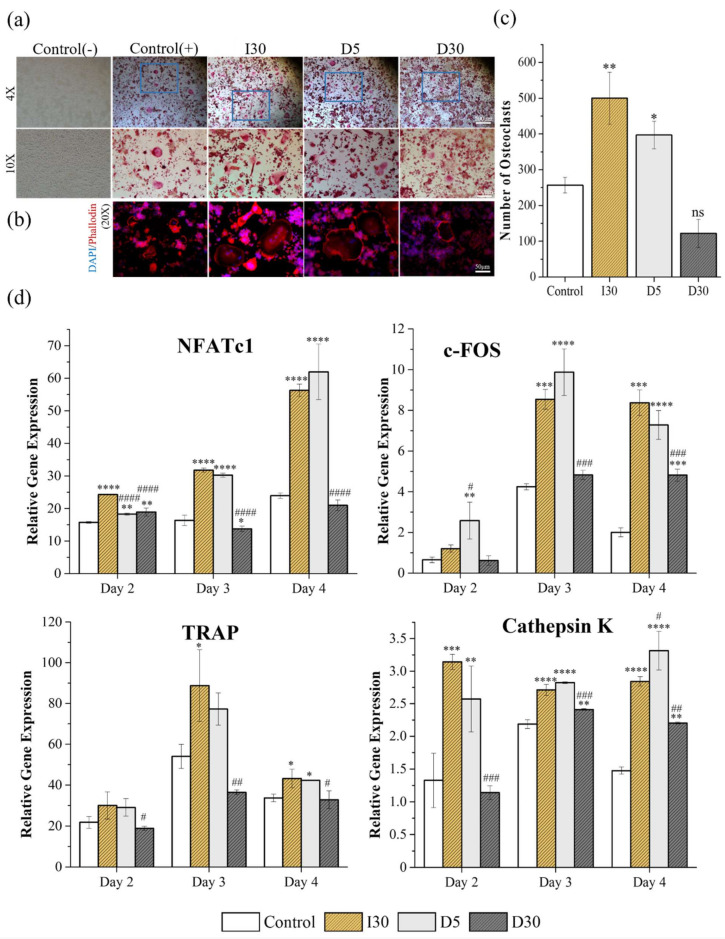
Osteoclast differentiation from the extract of NaI-incorporated paste (NaI@P) by (**a**) TRAP staining, and (**b**) Actin ring staining. (**c**) Number of osteoclast differentiation from TRAP staining. (**d**) The gene expression of c-FOS, NFATc1, cathepsin K, and TRAP measured by qPCR. The statistical significance of (**c**) was calculated using a one-way analysis of variance (ANOVA) and that of (**d**) was calculated using two-way ANOVA followed by a two-sided Tukey’s multiple comparison test. (*) compared to the positive control, (#) compared to the I30 group. */# represents *p* < 0.05, **/## *p* < 0.01, ***/### *p* < 0.001, ****/#### *p* < 0.0001, *n* = 3.

**Figure 5 molecules-27-02927-f005:**
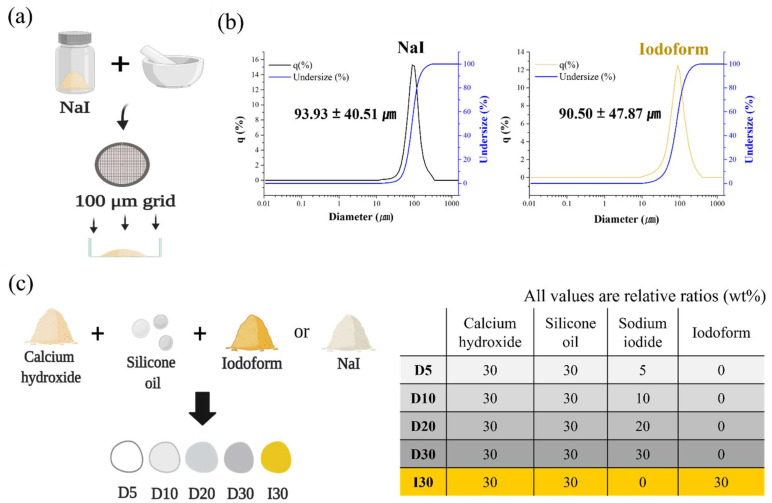
Development of NaI-incorporated paste (NaI@P). (**a**) Schematic illustration of the fabrication of strained NaI. (**b**) Graph comparing the particle size distribution of strained NaI and existing iodoform. (**c**) The composition of NaI-incorporated pastes (NaI@P) in a different proportions and iodoform-incorporated paste (Io@P).

**Table 1 molecules-27-02927-t001:** Flow, film thickness, viscosity at 0.5 Hz, and radiopacity of the samples (mean ± SD).

Samples	Flow (mm)	Film Thickness (mm)	Viscosity at 0.5 Hz (Pa∙s)	Radiopacity (mmAl)
D5	$26.67\pm 0.37{\;}^{a}$	$66.6\pm 8.45{\;}^{a}$	66.86	$2.24\pm 0.07{\;}^{a}$
D10	$26.20\pm 0.26{\;}^{a}$	$84.9\pm 7.68{\;}^{b}$	83.18	$3.41\pm 0.08{\;}^{b}$
D20	$24.23\pm 0.27{\;}^{b}$	$98.5\pm 3.95{\;}^{c}$	100.04	$5.66\pm 0.16{\;}^{c}$
D30	$24.23\pm 0.36{\;}^{b}$	$97.3\pm 4.79{\;}^{c}$	134.00	$6.37\pm 0.09{\;}^{d}$
I30	$19.11\pm 0.33{\;}^{c}$	$76.5\pm 7.76{\;}^{b}$	83.69	$7.45\pm 0.07{\;}^{e}$

Data with different superscript letters are significantly different.

**Table 2 molecules-27-02927-t002:** Water absorption and solubility of the samples (mean ± SD).

Samples	Water Absorption at 28 Days (g)	Solubility (%)	
		Solubility from Specimen	Solubility from Extract
D5	$1.50\pm 0.02{\;}^{a}$	$0.40\pm 0.08{\;}^{a}$	$1.83\pm 0.37^{\;a}$
D10	$1.78\pm 0.02{\;}^{b}$	$1.06\;\pm 0.12{\;}^{b}$	$2.59\pm 0.57{\;}^{a}$
D20	$2.31\pm 0.05{\;}^{c}$	$2.93\pm 0.17{\;}^{c}$	$3.24\pm 0.62{\;}^{a}$
D30	$2.34\pm 0.04{\;}^{c}$	$19.4\pm 0.21{\;}^{d}$	$6.25\pm 0.28{\;}^{b}$
I30	$1.01\pm 0.01{\;}^{d}$	$0.96\pm 0.12^{\;ab}$	$1.89\pm 0.28{\;}^{a}$

Data with different superscript letters are significantly different.

**Table 3 molecules-27-02927-t003:** Iodine, sodium, and calcium ion releases and pH of the samples (mean ± SD).

Samples	Ion Release (ppm)			pH
	Iodine	Sodium	Calcium	
I30	0	$3500.26\pm 29.58{\;}^{a}$	$61.18\pm 0.34{\;}^{a}$	$8.37\pm 0.12{\;}^{a}$
D5	393.6	$9838.65\pm 64.36^{\;b}$	$192.33\pm 0.37{\;}^{b}$	$8.40\pm 0.08{\;}^{b}$
D30	6719.8	$39,380.12\pm 109.62{\;}^{c}$	$525.04\pm 1.82{\;}^{c}$	$8.03\pm 0.05^{\;c}$

Data with different superscript letters are significantly different.

**Table 4 molecules-27-02927-t004:** Cell viability of the samples (mean ± SD).

Samples	Concentration of Samples			
	12.5%	25%	50%	100%
I30	89.10 ± 3.50 ***	97.32 ± 7.83	104.67 ± 7.20	110.73 ± 6.04 ***
D5	99.67 ± 5.79	97.55 ± 8.00	107.20 ± 8.80	117.14 ± 11.60 ****
D30	105.30 ± 10.56	107.20 ± 12.94	92.26 ± 7.70	77.13 ± 11.20 ***

*** *p* < 0.001; **** *p* < 0.0001 compared to the control (0% concentration).

**Table 5 molecules-27-02927-t005:** Number and size of differentiated osteoclasts by TRAP and actin ring staining (mean ± SD).

	Control	I30	D5	D30
Number of osteoclasts	257±27	$500\pm 89^{\;**}$	$397\pm 47{\;}^{*}\;$	122±48
Size of osteoclasts	10.24±0.60	$14.98\pm 0.63{\;}^{*}$	$12.76\pm 0.63{\;}^{\#}$	$11.24\pm 0.62{\;}^{\#\#}$

(*) compared to control group; (#) compared to I30. */# *p* < 0.05; **/## *p* < 0.01.

**Table 6 molecules-27-02927-t006:** Relative gene expression of osteoclast differentiation by q-PCR (mean ± SD).

Time	Primer	Relative Gene Expression			
		Control	I30	D5	D30
Day 2	NFATc1	15.70±0.25	$24.30\pm 0.16^{\;****}$	$18.30\pm 0.31^{\;**/\#\#\#\#}$	$18.91\pm 1.26^{\;**/\#\#\#\#}$
	c-Fos	0.66±0.13	1.20±0.18	$2.59\pm 0.91^{\;**/\#}$	0.62±0.23
	TRAP	21.80±2.90	30.10±6.60	29.11±4.30	$18.90\pm 1.00^{\;\#}$
	Cathepsin K	1.33±0.41	$3.14\pm 0.12^{\;***}$	$2.57\pm 0.50^{\;**}$	$1.14\pm 0.10^{\;\#\#\#}$
Day 3	NFATc1	16.40±1.60	$31.80\pm 0.64^{\;****}$	$30.24\pm 0.57^{\;****}$	$13.73\pm 0.86^{*/\#\#\#\#}$
	c-Fos	4.24±0.15	$8.54\pm 0.49^{\;***}$	$9.88\pm 1.10^{\;****}$	$4.83\pm 0.23^{\;\#\#\#}$
	TRAP	54.00±5.90	$88.70\pm 17.60^{\;*}$	77.2±8.00	$36.40\pm 1.20^{\;\#\#}$
	Cathepsin K	2.19±0.07	$2.71\pm 0.08^{\;***}$	$2.82\pm 0.01^{\;****}$	$2.41\pm 0.01^{\;**/\#\#\#}$
Day 4	NFATc1	23.95±0.87	$56.00\pm 1.92^{\;****}$	$61.97\pm 8.50^{\;****}$	$21.00\pm 1.63^{\;\#\#\#\#}$
	c-Fos	2.00±0.22	$8.37\pm 0.63^{\;***}$	$7.28\pm 0.70^{\;****}$	$4.80\pm 0.29^{\;***/\#\#\#}$
	TRAP	33.70±1.80	$43.20\pm 4.60^{\;*}$	$42.30\pm 0.10^{\;*}$	$32.80\pm 4.30^{\;\#}$
	Cathepsin K	1.47±0.05	$2.84\pm 0.07^{\;****}$	$3.32\pm 0.30^{\;****/\#}$	$2.20\pm 0.01^{\;**/\#\#}$

(*) compared to control group; (#) compared to I30. */# *p* < 0.05; **/## *p* < 0.01; ***/### *p* < 0.001; ****/#### *p* < 0.0001.

## Data Availability

Not applicable.

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
