# Peer review of "Improvement of Biological Effects of Root-Filling Materials for Primary Teeth by Incorporating Sodium Iodide"

_molecules, 2022, doi:10.3390/molecules27092927_

Round 1
Reviewer 1 Report
In this manuscript, different weight proportions of water-soluble iodide (NaI)-incorporated root-filling material were developed. Their physicochemical and biological properties were evaluated. The work is interesting, however, some revisions are needed.
(1) What about the degradation behavior of D30 or D5? It is important for a root-filling material.
(2) Do the DX materials have antibacterial activity or cytotoxicity for osteoblasts due to the released I? The release of I from D30 or D5 needs to be detected.
Author Response
Thank you for your comments and I revised and wrote point-by-point response, so please see the attachment, which contains revised manuscript, revised figures, and response to your comments.

Reviewer 2 Report
A thorough investigation was carried out by this work. However, its presentation needs to be improved.
The introduction is messy, the information doesn't flow as it should, there are distinct ideas displayed with no obvious link between them. It needs to be rewritten.
Line 57. Reference needed.
Line 84. Please find a link between the description of Endoflas and the data regarding root resorption (lines 76-84).
Line 90. "Therefore, Vitapex®-containing iodoform may hasten..." There is no obvious logic in presenting the information.
Line 99. Why didn't you use Vitapex instead of a home made version?
Materials and Methods should be section 2, followed by 3. results, 4. discussion etc.
The manufacturers date should be complete, including location.
By which means (based on what) did you choose the composition of the samples?
Line 401-405 belongs to subsection 4.2.
Line 408. "which was ground using a planetary ball mill (PM 100CM; Retsch Corporation) and strained through a 100 μm grid." Please remove, it just repeats previous information.
Same 410-411. "analyzed by the particle size analyzer. They had similar particle sizes of approximately 90 μm."
Line 411-413. Reformulate as caption, not as text. "(c) To fabricate NaI-incorporated paste, calcium hydroxide, silicone oil and strained NaI were blended in the proportion shown in Table, and the resulting materials were denoted D5, D10, D20, and D30. The control, called I30, was composed of calcium hydroxide, silicone oil, and iodoform in the same proportion."
Line 420, 423. Please replace we with third person. Same 428, 430 etc. Check the whole manuscript.
Line 436. Please give manufacturer's data for the rheometer.
Line 533. Which optical miroscope.
Table 1. The title should be brief, not repeat the data in it. Example: Physical properties of the samples.
Line 118. "Within each column, the significant differences between groups are indicated by different superscript letters." should be removed. Same as line 120.
Figure 1. Caption should be: Figure 1. Physical properties of NaI incorporated paste (NaI@P). (a) Flowability. (b) Film Thickness. (c) Viscosity (d) Radiopacity. (e) Water absorption by weight change over 28 days. The other information should be included in the main text, if necessary.
Same for Figure 2, Figure 3, Figure 4.
The results of the biological tests should be summarized in a table, for better understanding.
Lines 290-294. Should be moved to 4.3.1.
Line 295-299. Please rephrase.
Line 306, 308, 348, 362. By this do you mean your study? Please be clear and specific.
Line 382. "One of the limitations is that this study was an in vitro biological study". Please rephrase.
Line 568. As to my understanding, the comparison was not made to commercially available Vitapex, but to a homemade version. This is not a valid conclusion.
Please prepare the reference list according to MDPI template.
Author Response
Thank you for your comments and I revised and wrote point-by-point response to your comments, so please see the attachment.

Round 2
Reviewer 1 Report
The manuscript has been well revised and can be accepted.
Reviewer 2 Report
No further comments. Thank you for revising.